# Maternal serum levels of prokineticin-1 related to pregnancy complications and metformin use in women with polycystic ovary syndrome: a post hoc analysis of two prospective, randomised, placebo-controlled trials

Dorina Ujvari [1,2] Anastasia Trouva,[3] Angelica Lindén Hirschberg,[1,4] Eszter Vanky[5,6]

ALH and EV contributed equally.

For numbered affiliations see end of article.

**Correspondence to**
Dr Dorina Ujvari;
dorina.ujvari@ki.se

## ABSTRACT

**Objective** Serum prokineticin-1 (s-PROK1) in the second and third trimester of pregnancy is positively correlated to preeclampsia, intrauterine growth restriction (IUGR) and preterm delivery. Women with polycystic ovary syndrome (PCOS) are prone to these adverse pregnancy outcomes. However, the contribution of PROK1 to the development of pregnancy complications and the effect of metformin and hyperandrogenism on s-PROK1 in PCOS have not been studied previously.

**Design** This work is a post hoc analysis of two prospective, randomised, placebo-controlled trials.

**Setting** Pregnant women with PCOS were included from 11 study centres in Norway.

**Participants** From 313 women, 264 participated in the present study after exclusions due to dropouts or insufficient serum samples.

**Intervention** Women with PCOS were randomly administered with metformin or placebo, from first trimester to delivery.

**Primary and secondary outcome measures** s-PROK1 was analysed using ELISA at gestational week 19 and related to pregnancy complications, fasting insulin levels, homoeostatic model assessment for insulin resistance (HOMA-IR), testosterone, or androstenedione levels, metformin use, PCOS phenotype and hyperandrogenism.

**Results** Maternal s-PROK1 in the second trimester did not predict pregnancy-induced hypertension, pre-eclampsia or late miscarriage/preterm delivery in women with PCOS. However, s-PROK1 was lower in women who used metformin before inclusion, both in those randomised to metformin and to placebo, compared with those who did not. s-PROK1 was also lower in those who used metformin both at conception and during pregnancy compared with those who used metformin from inclusion or did not use metformin at all. s-PROK1 was lower in hyperandrogenic compared with normo-androgenic women with PCOS.

**Conclusions** Maternal s-PROK1 in the second trimester did not predict pregnancy complications in PCOS. Those who used metformin at conception and/or during pregnancy had lower s-PROK1. PCOS women with hyperandrogenism exhibited lower s-PROK1 compared with normo-adrogenic phenotypes.

**Trial registration number** NCT03259919 and NCT00159536.

## STRENGTHS AND LIMITATIONS OF THIS STUDY

⇒ This study is a post hoc analysis of two randomised, double-blinded, placebo-controlled, multi-centre trials.
⇒ The study includes 264 well-characterised pregnant women with PCOS.
⇒ The number of pregnancy complications was limited, potentially leading to type II errors.
⇒ Due to the broad timeframe of sampling in the first trimester and the lack of predictive value at week 32 and 36, s-PROK1 was measured only at week 19.

## INTRODUCTION

Polycystic ovary syndrome (PCOS) is the most common endocrine disorder in women at their reproductive age. Its prevalence is 10–17% based on at least two of the three features of PCOS (oligo-amenorrhea, hyperandrogenism and polycystic ovaries) according to the Rotterdam criteria.[1] Approximately 70% of these patients have insulin resistance and compensatory hyperinsulinaemia.[2] PCOS is closely related to subfertility and pregnancy complications, including miscarriage, gestational diabetes, preeclampsia, intrauterine growth restriction (IUGR) and preterm delivery.[3 4] Hyperinsulinaemia and hyperandrogenism might contribute to the development of these complications.[5 6]

Prokineticin-1 (PROK1) is a pleiotropic factor secreted by endocrine glands, with a broad range of biological functions. In the

female reproductive system, it is expressed in the endometrium, ovaries, adrenal glands and placenta[7 8] and seems to be a pivotal actor at all the different stages of the gestation, from follicular development to parturition. PROK1 is significantly increased in first trimester decidua, compared with non-pregnant endometrium and controls trophoblast migration and invasion into the decidua and angiogenesis in the developing placenta.[9 10] The placenta is thought to be the main source of PROK1 in the maternal circulation.[11 12] In uncomplicated pregnancies, serum level of PROK1 (s-PROK1) increases fivefold in the first trimester compared with non-pregnant state[13] and promotes the proliferation of anchoring cytotrophoblasts and formation of trophoblastic plugs in the maternal spiral arteries.[11 14] After gestational week 10–12, PROK1 levels decline, facilitating the transformation of trophoblasts into an invasive phenotype, the remodelling of the maternal spiral arteries and the increase of placental oxygen tension to promote placental angiogenesis in order to supply the growing fetus.[11 13] Excess expression of PROK1 was demonstrated to cause poor invasion of extravillous trophoblasts in the first trimester. This potentially leads to imperfect trophoblastic plugs resulting in early pregnancy loss, or in less severe cases to insufficient remodelling of maternal spiral arteries and consequently to impaired placental development. The relatively hypoxic milieu in the abnormal placenta leads to the upregulation of several hypoxia-regulated factors, among others PROK1.[10] Dysregulation of PROK1 is associated with preeclampsia, IUGR and preterm delivery.[13–16] We have previously shown that insulin upregulates PROK1 in decidualising endometrial stromal cells in vitro, and androgens potentiate this enhancement.[17 18] Despite hyperinsulinaemia/hyperandrogenism and an increased occurrence of these pregnancy complications, the role of PROK1 has not been studied in PCOS.

The administration of metformin, an insulin sensitiser, in the treatment of non-pregnant PCOS women is well established.[19] Evidence shows its usefulness in reducing late miscarriages and preterm deliveries; however, its efficacy on other complications is debated and the mechanism of action is not fully understood.[20–23]

The primary objective of this study was to investigate if s-PROK1 predicts pregnancy-induced hypertension, preeclampsia and late miscarriage/preterm delivery. We hypothesised that metformin treatment was associated with lower, and hyperandrogenism with higher s-PROK1 in the second trimester.

## MATERIALS AND METHODS
### Study design
This work is a post hoc analysis of two prospective, randomised, placebo-controlled trials, investigating the effect of metformin on pregnancy complications in women with PCOS. 264 women participated with analysed serum samples in the present study, 128 and 136 in the metformin and placebo group, respectively (figure 1).

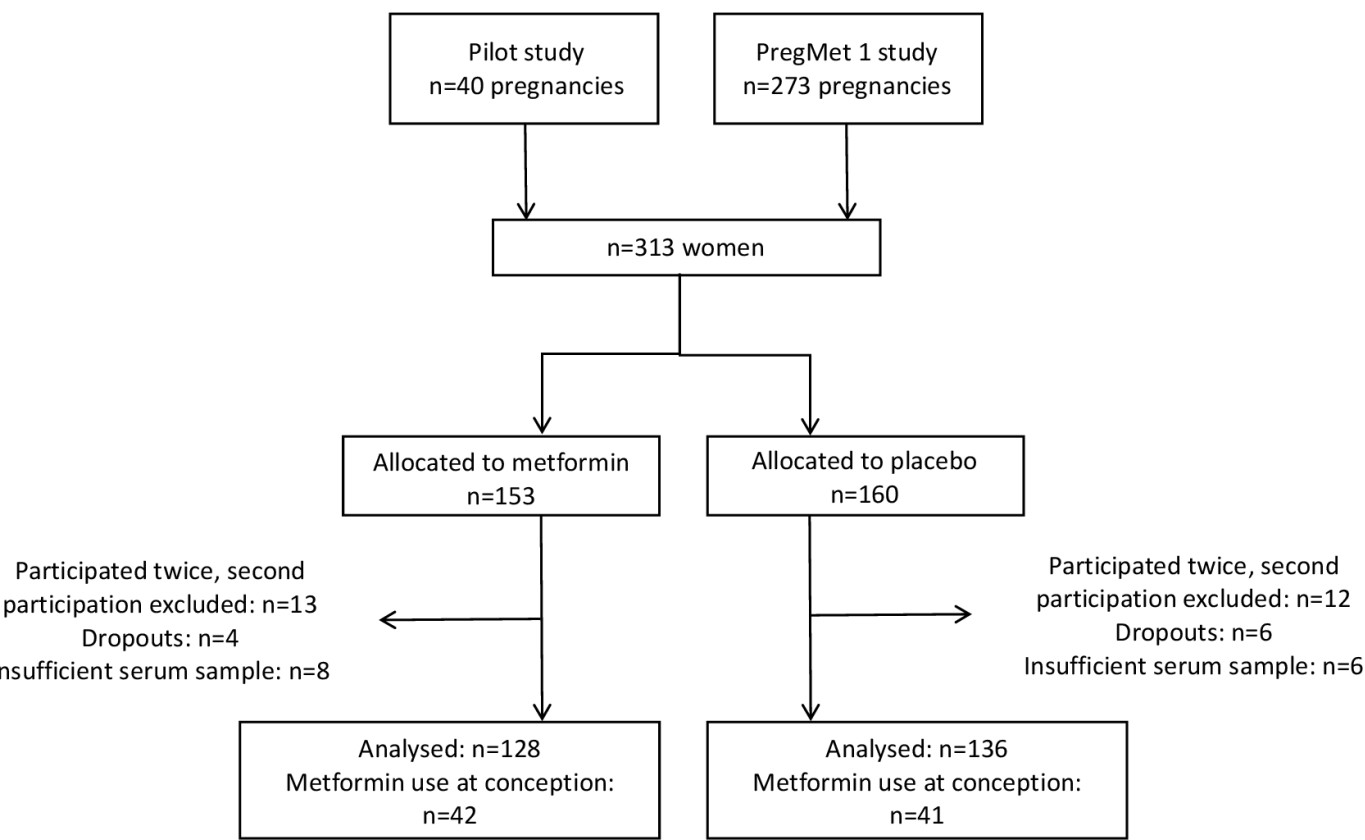

**Figure 1** Flowchart of the original and the secondary studies.

The clinical studies[20 21] are described in the supplementary material.

## Enzyme-linked immunosorbent assay

Serum samples, collected at gestational week 19, were analysed by a PROK1 ELISA kit (Thermo Fischer Scientific), in 1:4 dilution. Absorbance of the samples was measured in duplicates at 450 and 550 nm on a Multiscan FC photometer, and $A_{450}$-$A_{550}$ values were used to correct for optical imperfections in the microplate. To plot the standard curves, the four-parameter logistic curve fit was applied. The limit of detection (LOD) was 13.72 pg/mL. Values below the detection limit were replaced by 0.5xLOD. The intra-assay and inter-assay coefficients of variations were <10% and <12%, respectively.

## Patient and public involvement

None.

## Statistical analysis

Statistical procedures were performed using SPSS by IBM V.25 and Statistica 14.0 TIBCO software. Differences in baseline characteristics were analysed using unpaired t-test and Fischer's exact test. s-PROK1 was positively skewed and therefore log-transformed. Logistic regression analysis was used to analyse whether s-PROK1 predicts pregnancy complications. Differences of s-PROK1 between subgroups were analysed by median test, followed by pairwise median test with Bonferroni correction, in case of statistical significance. Correlations between s-PROK1 and other variables were evaluated using Spearman's rank order correlation test. $p < 0.05$ was considered as statistically significant.

## RESULTS

### Study population

Baseline characteristics of the women in the metformin and the placebo groups were comparable (online supplemental table 1). Baseline characteristics were compared in additional subgroups (online supplemental table 1). Weight, body mass index (BMI), fasting insulin levels and homoeostatic model assessment for insulin resistance (HOMA-IR) were higher in hyperandrogenic compared with normo-androgenic participants (p=0.034, p=0.015, p=0.029 and p=0.034, respectively). The mode of conception was significantly different between subgroups used and not used metformin during conception (p=0.003) and between women who took metformin both at conception and during pregnancy compared with those who did not use metformin at all (p=0.048).

### Pregnancy complications

Pregnancy complications are listed in online supplemental table 2. s-PROK1 did not predict pregnancy-induced hypertension (n=15) (OR 0.95, 95% CI 0.32 to 2.74, p=0.92), preeclampsia (n=15) (OR 0.47, 95% CI 0.14 to 1.35, p=0.17) or late miscarriage/preterm delivery (n=20) (OR 0.62, 95% CI 0.24 to 1.56, p=0.32).

### PROK1 and metformin use at conception, during pregnancy, and both at conception and during pregnancy

s-PROK1 was lower in women who used metformin at conception (median=6.86, 95% CI 6.86 to 11.22) compared with those who did not (median=19.53, 95% CI 7.00 to 30.83) (p=0.012), regardless of later randomisation to metformin or placebo (figure 2A). s-PROK1

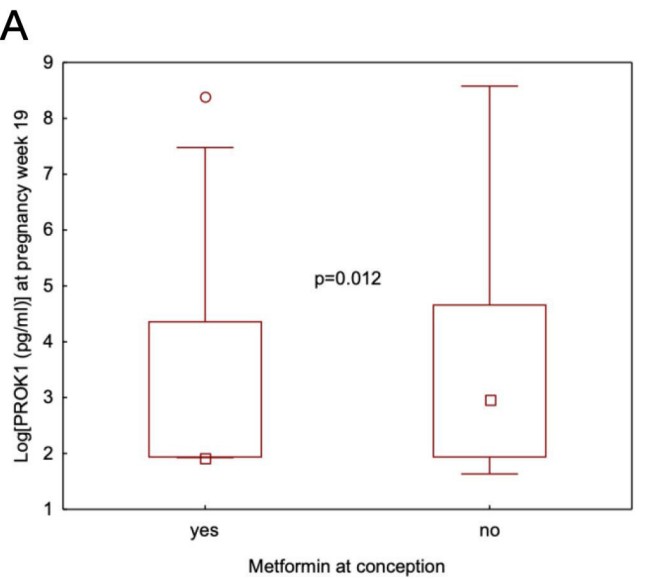
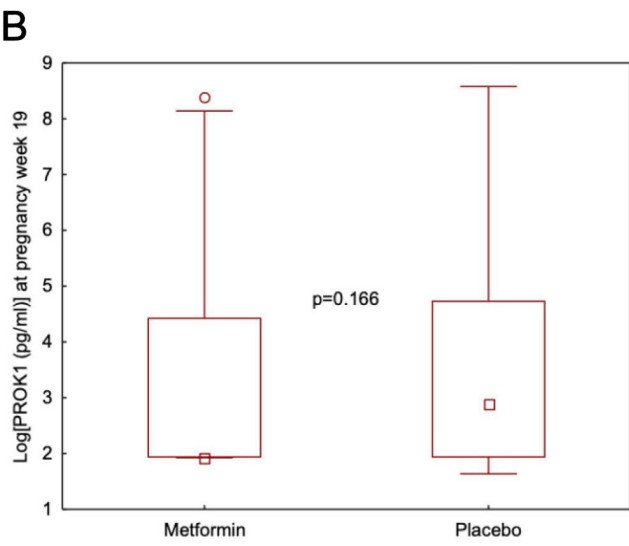

**Figure 2** (A) Serum levels of PROK1 at pregnancy week 19 in women with PCOS, who took metformin at conception (n=83) compared with those, who did not use metformin at conception (n=181). Values are presented as median, interquartile ranges (P25–P75) and non-outlier ranges (min-max). Open circles represent outliers (1.5*the box-height from the box). (B) Serum levels of PROK1 at pregnancy week 19 in women with PCOS, who took metformin (n=128) compared with those, who took placebo after randomisation (n=136). Values are presented as median, interquartile ranges (P25–P75) and non-outlier ranges (min-max). Open circles represent outliers (1.5*the box-height from the box).

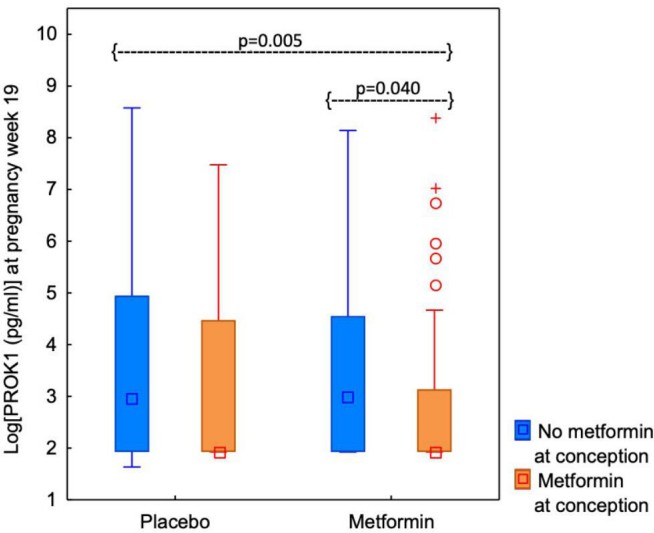

**Figure 3** Serum levels of PROK1 at pregnancy week 19 in women with PCOS, who did not use metformin/used metformin at conception in the placebo group (n=41 and n=95, respectively) and in the metformin group (n=42 and n=86, respectively). Values are presented as median, interquartile ranges (P25–P75) and non-outlier ranges (min-max). Open circles represent outliers (1.5*the box-height from the box) and the stars represent extremes (3*the box-height from the box).

was not different in women randomised to metformin (median=6.86, 95% CI 6.86 to 20.51) or placebo (median=17.75, 95% CI 7.00 to 30.83) during pregnancy (p=0.166) (figure 2B). Median test showed a statistically significant difference between the four subgroups of no metformin use/metformin use at conception combined with metformin/placebo use during pregnancy (p=0.020). Pairwise median test revealed that s-PROK1 was lower in women who used metformin both at conception and during pregnancy (median=6.86, 95% CI 6.86 to 18.00), compared with those who used metformin during pregnancy (median=19.76, 95% CI 9.60 to 46.46) (p=0.008, $p_{adjusted}$=0.040) (figure 3). s-PROK1 was lower in women who took metformin both at conception and during pregnancy compared with those who did not use metformin at all (median=19.53, 95% CI 7.00 to 40.36) (p=0.001, $p_{adjusted}$=0.005) (figure 3). The difference between those who used metformin at conception and during pregnancy compared with those who used metformin at conception and placebo during pregnancy (median=6.86, 95% CI 6.86 to 68.70) did not reach the level of statistical significance (p=0.060, $p_{adjusted}$=0.301) (figure 3). There was no statistically significant difference between women who did not use compared with those who used metformin at conception in the placebo group (p=0.577, $p_{adjusted}$=1) and between women who did not use metformin at conception, but took metformin compared with women who took placebo during pregnancy (p=0.832, $p_{adjusted}$=1) (figure 3).

## PCOS phenotypes
Differences in s-PROK1 were also analysed in subgroups stratified according to PCOS phenotypes. Patients from

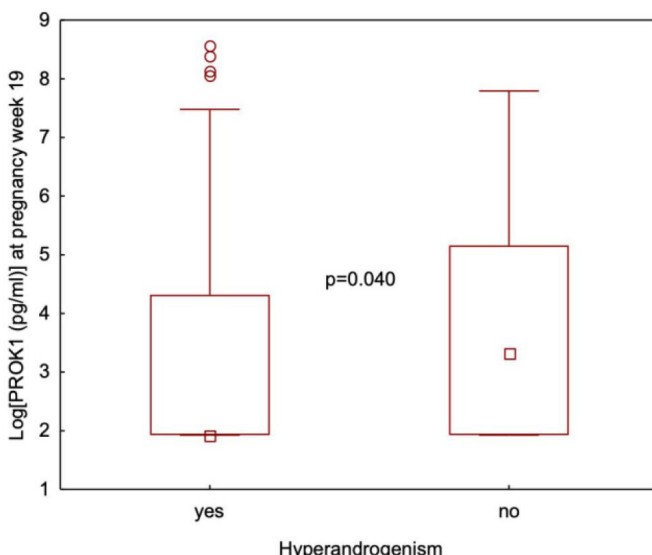

**Figure 4** Serum levels of PROK1 at pregnancy week 19 in women with PCOS and hyperandrogenism (n=164) compared with those without hyperandrogenism (n=64). Values are presented as median, interquartile ranges (P25–P75) and non-outlier ranges (min-max). Open circles represent outliers (1.5*the box-height from the box).

the pilot study (n=37) were not included in this analysis, as they were diagnosed according to the NIH criteria, and due to the lack of data in the presence or absence of polycystic ovaries, those fulfilling all three criteria and hyperandrogenism-oligo-amenorrhea groups could not been distinguished. In PregMet1 samples, s-PROK1 was not significantly different between PCOS phenotypes A (HA+PCO+OA), B (HA+PCO), C (HA+OA) and D (PCO+OA) (p=0.177) (data not shown). s-PROK1 was lower in hyperandrogenic (median=6.86, 95% CI 6.86 to 9.69) compared with normo-androgenic women (median=6.86, 95% CI 7.00 to 88.69) (p=0.040) (figure 4).

## CORRELATIONS
s-PROK1 showed no correlation with BMI, fasting insulin levels, HOMA-IR, testosterone, or androstenedione levels either at inclusion or at week 19 of gestation (online supplemental table 3).

## DISCUSSION
The primary aim of this study was to investigate if s-PROK1 predicts pregnancy-induced hypertension, preeclampsia and late miscarriage/preterm delivery in women with PCOS. Previously, s-PROK1 was reported to be higher in the second and third trimesters in preeclamptic and IUGR pregnancies compared with uncomplicated pregnancies.[13 14] We and others have also demonstrated that PROK1 restricts the migration and invasion of trophoblast cells in vitro, suggesting that elevated s-PROK1 impairs embryo implantation and placentation.[13 18] However, s-PROK1 in the second trimester did not predict pregnancy-induced hypertension or preeclampsia in

our women with PCOS. It has also been demonstrated that PROK1 is upregulated in the myometrium and the placenta during labour, suggesting a role for PROK1 in the induction of term and preterm labour.[15 16] However, in this study, s-PROK1 did not predict late miscarriages/ preterm deliveries in pregnant women with PCOS, but we cannot exclude that a limited number of cases might have influenced our results.

We also investigated the impact of metformin on s-PROK1 and found lower levels by metformin use at conception, both with and without metformin use after inclusion, suggesting that metformin intake, especially in early pregnancy, might have a lowering effect on s-PROK1. However, participants were not randomised at this time-point and the lower s-PROK1 at week 19 might reflect that those who used metformin at conception differed from those who did not. It is known that metformin decreases insulin levels, although the treatment does not seem to prevent gestational diabetes in PCOS women.[20 21 23] We recently demonstrated that insulin upregulates PROK1 in decidualising endometrial stromal cells in vitro.[18] However, in this study, we found no correlation between s-PROK1 and insulin levels.

A possible association between s-PROK1 and hyper-androgenism was also investigated. In contrast to our hypothesis, s-PROK1 was lower in women with hyperan-drogenic phenotypes compared with normo-androgenic phenotype. However, there was no correlation between s-PROK1 and androgen levels at inclusion and week 19. Direct and indirect evidence support the role of andro-gens in the pathogenesis of placenta-related pregnancy complications.[24] We have previously showed that andro-gens potentiate insulin to upregulate PROK1 in decid-ualising endometrial stromal cells in vitro.[17] Increased maternal serum androgens have been reported in preeclamptic pregnancies.[25–28] PCOS women with preg-nancy complications have higher androgen levels than those without.[29] It was also shown that trophoblast inva-sion and placentation are impaired in PCOS compared with controls, and this impairment is associated with insulin resistance and circulating testosterone concen-trations.[30] However, androgens do not seem to directly contribute to the regulation of s-PROK1, although hyper-androgenism could be indirectly involved.

### Strengths and limitations of this study

There are several strengths of the present study including the well-characterised women with PCOS and the randomised, double-blinded placebo-controlled multi-centre study design, prospective data sampling and metic-ulous quality-checked diagnosis control. Some limitations should also be mentioned. There was a slight difference between the two trials regarding the metformin dose, 1700 mg and 2000 mg daily, respectively, which might have affected our results. Since maternal s-PROK1 has a dynamic course during pregnancy, timing of blood sampling for determination of s-PROK1 is crucial. Based on previous studies showing elevated s-PROK1 in the

second and third trimester of preeclamptic and IUGR pregnancies, we chose to analyse s-PROK1 in gestational week 19, representing the second trimester. We did not measure the s-PROK1 in serum samples taken at inclu-sion (between gestational week 5 and 12), as s-PROK1 has been shown to decline remarkably within this period and patients could not have been comparable. s-PROK1 was not measured at gestational week 32 and 36 either, as these late timepoints would not hold predictive values for gestational diabetes, early onset preeclampsia or late miscarriage. However, we cannot exclude that s-PROK1 should have been determined at several timepoints to study the course of PROK1 for pregnancy complications in PCOS. Furthermore, the number of pregnancy compli-cation cases was limited possibly leading to type II errors.

### Conclusions

S-PROK1, in the second trimester, did not predict pregnancy-induced hypertension, pre-eclampsia, or late miscarriage/preterm delivery in pregnant women with PCOS. s-PROK1 was lower in women who used metformin at conception. Women who used metformin at concep-tion and during pregnancy compared with those who used metformin from inclusion or not at all and those with hyperandrogenism compared with normo-androgenism exhibited lower s-PROK1. Further studies are needed to determine the exact role of s-PROK1 in pregnancy complications and the involvement of metformin and hyperandrogenism in the regulation of PROK1 in PCOS.

**Author affiliations**
[1]Department of Women's and Children's Health, Karolinska Institute, Stockholm, Sweden
[2]Department of Microbiology, Tumor and Cell Biology; National Pandemic Centre; Centre for Translational Microbiome Research, Karolinska Institute, Solna, Sweden
[3]Department of Molecular Medicine and Surgery, Karolinska Institute, Stockholm, Sweden
[4]Department of Gynecology and Reproductive Medicine, Karolinska University Hospital, Stockholm, Sweden
[5]Department of Clinical and Molecular Medicine, Norwegian University of Science and Technology, Trondheim, Norway
[6]Department of Gynaecology and Obstetrics, St Olav's University Hospital, Trondheim, Norway

**Acknowledgements** The authors thank Weifa A/S for delivering metformin and placebo tablets free of charge.

**Contributors** EV conceived and designed the original protocol. On the initiative of DU and ALH, the protocol of the present study was written together with EV. DU run the analysis, DU and AT managed the database and analysed the data with the input from EV and ALH. A statistician performed the statistical analyses. DU and AT drafted the manuscript with input from ALH, EV. DU is responsible for the overall content as guarantor. All the authors contributed to the article, reviewed and approved the submitted version.

**Funding** This work was supported by grants from the Liaison Committee between the Central Norway Regional Health Authority and the Norwegian University of Science and Technology, Swedish Research Council (2021-01348), Stockholm County Council (2019-0248) and Åke Wiberg foundation (M18-0114).

**Competing interests** None declared.

**Patient and public involvement** Patients and/or the public were not involved in the design, or conduct, or reporting, or dissemination plans of this research.

**Patient consent for publication** Not applicable.

**Ethics approval** An informed consent was obtained from each patient and the Declaration of Helsinki was followed throughout the studies. The Committee for Medical Research Ethics of Health Region IV, Norway and The Norwegian Medicines Agency approved the studies (project numbers: 51–2000 and 145.04) and the supplement application for the current study (reference number 2010/3115 (145-04)). The clinical trials were registered at www.clinicaltrials.gov as NCT03259919 (The pilot study) and NCT00159536 (The PregMet1 study).

**Provenance and peer review** Not commissioned; externally peer reviewed.

**Data availability statement** Data are available upon reasonable request.

**ORCID iD**
Dorina Ujvari http://orcid.org/0000-0002-4659-9712

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
