## [Reviewer comments · BMJ Open]

ARTICLE DETAILS

TITLE (PROVISIONAL)	Maternal serum levels of prokineticin-1 related to pregnancy complications and metformin use in women with polycystic ovary syndrome: a post-hoc analysis of two prospective, randomized, placebo-controlled trials
AUTHORS	Ujvari, Dorina; Trouva, Anastasia; Linden, Angelica; Vanky, Eszter

VERSION 1 – REVIEW

REVIEWER	Palomba, Stefano Univ Modena
REVIEW RETURNED	28-Apr-2023

GENERAL COMMENTS	Title: Maternal serum levels of prokineticin-1 related to pregnancy complications and metformin use in women with polycystic ovary syndrome Manuscript ID: bmjopen-2023-073619 General comments The current is a post-hoc analysis of two RCTs and was designed to study the contribution of PROK1 to the development of pregnancy complications in 264 women with PCOS, and the effect of metformin and hyperandrogenism on this marker. Serum PROK1 in the second trimester did not predict main pregnancy complications in PCOS, even if its levels were lower in women who used metformin pretreatment with special regard for women randomized to receive metformin and in hyperandrogenic subjects with PCOS. The study is interesting. However, it is lacking control (non-PCOS) group. I suggest to carefully revise the paper according to the comments below reported. Specific comments Have you assessed marker of low-grade chronic inflammation (doi: 10.1210/jc.2014-1214) and their relationship with PROK1? Data on PCOS phenotypes should be complete. I suggest subcategorizing your population in the four main phenotypes. Data on pregnancies should be given. For example, were the pregnancies natural, obtained after ovulation induction (drug? doses?) or after IVF treatments? The authors should better explain the mechanism underling the pregnancy complication in PCOS. Only in this way they may explain why PROK1 levers are not predictor of pregnancy complications in PCOS (contrarily to data on general population). To this regard, several points are not adequately discussed. 1. the role of endometrium (doi: 10.1093/humupd/dmaa051) and potentially of oocyte/embryos (doi: 10.1016/j.tem.2016.11.008) are determinants for pregnancy complications in PCOS; 2. the increased risk in
---

	pregnancy complications (doi: 10.1093/humupd/dmv029) in women with PCOS is essentially due to abnormal trophoblast invasion (doi: 10.1210/jc.2012-1100) and placentation (doi: 10.1093/humrep/det250), and these alterations are closely related to PCOS phenotypes in terms of histological findings (doi: 10.1016/j.rbmo.2014.04.010) and of clinical outcomes (doi: 10.1016/j.fertnstert.2009.10.043).
--	---

REVIEWER	Ng, Ernest The University of Hong Kong, Department of Obstetrics and Gynecology
REVIEW RETURNED	30-Apr-2023

GENERAL COMMENTS	This study is a post-hoc analysis of two randomized placebo-controlled trials investigating the effect of metformin on pregnancy complications in women with PCOS. It was shown that metformin given from first trimester to delivery did not reduce pregnancy complications in PCOS including preeclampsia and gestational diabetes. The aim of this study is to investigate if s-PROK1 predicts pregnancy induced hypertension, preeclampsia, and late miscarriage/ preterm delivery. Strength of the study  1. A multi-centre study 2. Double blind trial Major comments  1. Introduction: The hypothesis was metformin treatment was associated with lower, and hyperandrogenism with higher s-PROK1 in the second trimester. (P5L3-5) What is the basis for such hypothesis? How is this linked to the aim of the study? 2. Materials and method: Blood samples were taken from women at the gestational age 19 weeks. (P5L26) It is not stated why the blood samples at gestational age 19 weeks were measured to predict the risk of the pregnancy complications. How will the gestational weeks affect the s-PROK1 levels especially in the second trimester? 3. Materials and method: Women randomized to the metformin group had received metformin from the first trimester, which may affect the measurement of s-PROK1 levels in the second trimester. Should the authors restrict the analysis to the placebo group investigating if s-PROK1 predicts the pregnancy complications? Then the metformin group would be investigated to evaluate the effect of metformin on serum s-PROK1 levels. 4. Materials and method: Although this is a post-hoc analysis of two randomized placebo-controlled trials, the authors did not state if the sample size was large enough to detect difference.
--

REVIEWER	Rasmussen, Svein Universitetet i Bergen
REVIEW RETURNED	24-May-2023

GENERAL COMMENTS	The study is original and well conducted, although in have some limitations, which however, are acknowledged. I think it may be published as it is.
---

REVIEWER	Hurme, Saija University of Turku, Department of Biostatistics
REVIEW RETURNED	20-Jun-2023

GENERAL COMMENTS	 1. How was the sample size defined? Is there any sample size calculations performed when the study was planned? 2. Patients were randomized to metformin or placebo group and stratified according to metformin use at conception. Even though the patients using metformin before had wash out -period before the “intervention”, metformin use at conception seems to be quite important confounding factor. Because of that I believe it should be accounted in the analyses using more complicated models than Mann-Whitney U-test. If the distribution of S-PROK1 is normal whit log-transformation, then the effect of metformin could be studied using two-way ANOVA where randomized group, metformin use at conception and the interaction of those are included and the effect of different things could be studied better than when different versions of groups are tested separately as is done now. 3. I think also the effect of study should be accounted in the analyses especially when the intervention was not exactly the same in both studies. 4. What was pilot study made for? Is there some information that was used in the planning of PregMet -study or what was the idea of pilot study? 5. Statistical analyses: Where is log-transformed S-PROK1 used? Analyses were reported to be performed using non-parametric tests where normality is not assumed. 6. Results and tables: Exact p-values should be presented instead of only below or over 0.05. 7. Results presented in Figures 2 and 3 should be presented also more accurately with exact numbers. It is difficult to evaluate the difference between groups without exact values (with 95% confidence intervals if applicable). Also the clinical importance of difference could be discussed. 8. Figure 1: Metformin use at conception would be good additional information in flow chart. 9. Figure 2: It would be more informative to see one figure with four groups according to randomized group and metformin use at conception. 10. Figures 2 and 3: Number of observations in each group should be presented in the figures.
---

VERSION 1 – AUTHOR RESPONSE

Reviewer: 1

Dr. Stefano Palomba, Univ Modena

Comments to the Author:

Title: Maternal serum levels of prokineticin-1 related to pregnancy complications and metformin use in women with polycystic ovary syndrome

Manuscript ID: bmjopen-2023-073619

General comments

The current is a post-hoc analysis of two RCTs and was designed to study the contribution of PROK1 to the development of pregnancy complications in 264 women with PCOS, and the effect of metformin and hyperandrogenism on this marker. Serum PROK1 in the second trimester did not predict main pregnancy complications in PCOS, even if its levels were lower in women who used metformin pretreatment with special regard for women randomized to receive metformin and in hyperandrogenic subjects with PCOS.

The study is interesting. However, it is lacking control (non-PCOS) group. I suggest to carefully revise the paper according to the comments below reported.

Specific comments

Have you assessed marker of low-grade chronic inflammation (doi: 10.1210/jc.2014-1214) and their relationship with PROK1?

We did not aim to analyze the relationship between low-grade inflammation and s-PROK1. However, C-reactive protein was measured in these pregnant women for another study, so we could not analyze a possible association. Spearman rank correlation test showed no significant association between hsCRP and s-PROK1 at gestational week 19 ($R=0.055$, $p=0.379$).

Data on PCOS phenotypes should be complete. I suggest subcategorizing your population in the four main phenotypes.

We included the results on the four main PCOS phenotypes in the manuscript and in Supplementary table 1, too, as suggested by the reviewer.

Data on pregnancies should be given. For example, were the pregnancies natural, obtained after ovulation induction (drug? doses?) or after IVF treatments?

Available data on the mode of conception (spontaneous, clomiphene citrate, IVF/ICSI and other) is added now to the Supplementary table 1.

The authors should better explain the mechanism underlying the pregnancy complication in PCOS. Only in this way they may explain why PROK1 levels are not predictor of pregnancy complications in PCOS (contrarily to data on general population). To this regard, several points are not adequately discussed.

1. the role of endometrium (doi: 10.1093/humupd/dmaa051) and potentially of oocyte/embryos (doi: 10.1016/j.tem.2016.11.008) are determinants for pregnancy complications in PCOS;
2. the increased risk in pregnancy complications (doi: 10.1093/humupd/dmv029) in women with PCOS is essentially due to abnormal trophoblast invasion (doi: 10.1210/jc.2012-1100) and placentation (doi: 10.1093/humrep/det250), and these alterations are closely related to PCOS phenotypes in terms of histological findings (doi: 10.1016/j.rbmo.2014.04.010) and of clinical outcomes (doi: 10.1016/j.fertnstert.2009.10.043).

We agree with the reviewer that the underlying reasons for pregnancy complications in PCOS were poorly explained. We also agree that the role of dysregulated decidualization due to insulin resistance, hyperandrogenemia or other contributing factors, the quality of the embryo, the insufficient maternal spiral artery remodeling and impaired placentation are all reasons for increased incidence of pregnancy complications in PCOS women. However, we believe that the impaired invasion of trophoblastic cells into the maternal decidua, the impaired maternal spiral artery remodeling and the abnormal placental development are direct results of dysregulated decidualization and modulated secretome of the endometrium in PCOS women. As we showed in earlier studies, that PROK1 is highly up-regulated by insulin (DOI: 10.1111/jcmm.13305) and the action of insulin is enhanced by the presence of androgens (DOI: 10.1111/jcmm.14923) in decidualizing stromal cells, we aimed to investigate if the serum level of PROK1 is associated with pregnancy complications in women with PCOS. Thus, we do not hypothesize that the mechanism is different in PCOS women compared to non-PCOS women, only that PCOS women, due to dysregulated decidual secretome, are more prone to develop these pregnancy complications. We agree that it was misleadingly phrased in the manuscript. We reformulated this part and added more information regarding the mechanism to the Introduction.

Reviewer: 2

Dr. Ernest Ng, The University of Hong Kong

Comments to the Author:

This study is a post-hoc analysis of two randomized placebo-controlled trials investigating the effect of metformin on pregnancy complications in women with PCOS. It was shown that metformin given from first trimester to delivery did not reduce pregnancy complications in PCOS including preeclampsia and gestational diabetes. The aim of this study is to investigate if s-PROK1 predicts pregnancy induced hypertension, preeclampsia, and late miscarriage/ preterm delivery.

Strength of the study

1. A multi-centre study
2. Double blind trial

Major comments

1. Introduction: The hypothesis was metformin treatment was associated with lower, and hyperandrogenism with higher s-PROK1 in the second trimester. (P5L3-5) What is the basis for such hypothesis? How is this linked to the aim of the study?

We thank the reviewer for noticing this deficiency. Indeed, we wrote about the possible link between hyperinsulinemia/hyperandrogenism and PROK1 in the Discussion, but this information was missing from the Introduction. We completed the Introduction with this information.

2. Materials and method: Blood samples were taken from women at the gestational age 19 weeks. (P5L26) It is not stated why the blood samples at gestational age 19 weeks were measured to predict the risk of the pregnancy complications. How will the gestational weeks affect the s-PROK1 levels especially in the second trimester?

We collected blood samples from the pregnant women at inclusion (between week 5-12), week 19, 32 and 36. As it is established in the literature, PROK1 is relatively high in the first trimester and declines after week 10-12 to facilitate the differentiation of trophoblasts into an invasive phenotype, the remodeling of the maternal spiral arteries and the increase of placental oxygen tension to promote placental angiogenesis. We hypothesized that second trimester blood sample would reveal possible dysregulation of PROK1. We have not considered measuring PROK1 in serum samples taken at inclusion as they were not uniform, given that samples were taken somewhere between gestational week 5 and 12, and serum level of PROK1 is declining remarkably within this period. Furthermore, measurement of PROK1 in serum samples at gestational week 32 and 36 would not have predictive values timewise in cases of late miscarriage, gestational diabetes and early onset preeclampsia.

3. Materials and method: Women randomized to the metformin group had received metformin from the first trimester, which may affect the measurement of s-PROK1 levels in the second trimester. Should the authors restrict the analysis to the placebo group investigating if s-PROK1 predicts the pregnancy complications? Then the metformin group would be investigated to evaluate the effect of metformin on serum s-PROK1 levels.

In a previous publication it was shown that there was no difference in the prevalence of gestational diabetes and pre-eclampsia between the metformin and the placebo group in an epi-analysis of these two randomized, placebo-controlled trials (DOI: 10.1111/aogs.12015). Furthermore, the number of pre-eclampsia and gestational diabetes cases were limited in this study and investigating separately the placebo-group would have decreased the number of cases even more. For these reasons we chose to analyze the metformin and placebo groups together. To investigate the effect of metformin on s-PROK1 we needed the placebo group as a control group.

4. Materials and method: Although this is a post-hoc analysis of two randomized placebo-controlled trials, the authors did not state if the sample size was large enough to detect difference.

Due to the lack of a pilot study with a similar setup in this patient group, and because it is a post-hoc analysis of two randomized placebo-controlled trials, sample size estimation could not be performed. We are aware of that the number of pregnancy complication cases was limited possibly leading to type II errors and stated it as a limitation.

Reviewer: 3

Dr. Svein Rasmussen, Universitetet i Bergen

Comments to the Author:

The study is original and well conducted, although it has some limitations, which however, are acknowledged. I think it may be published as it is.

Reviewer: 4

Dr. Saija Hurme, University of Turku

Comments to the Author:

1. How was the sample size defined? Is there any sample size calculations performed when the study was planned?

Due to the lack of a pilot study with a similar setup in this patient group to measure s-PROK1, and because it is a post-hoc analysis of two randomized placebo-controlled trials, originally designed to measure the effect of metformin on androgen levels in pregnant women with PCOS, and to investigate the prevalence of preterm deliveries and insulin-requiring gestational diabetes, sample size estimation could not be performed. We are aware of that the number of pregnancy complication cases was limited, possibly leading to type II errors, and stated it as a limitation.

2. Patients were randomized to metformin or placebo group and stratified according to metformin use at conception. Even though the patients using metformin before had wash out -period before the "intervention", metformin use at conception seems to be quite important confounding factor. Because of that I believe it should be accounted in the analyses using more complicated models than Mann-Whitney U-test. If the distribution of S-PROK1 is normal whit log-transformation, then the effect of metformin could be studied using two-way ANOVA where randomized group, metformin use at conception and the interaction of those are included and the effect of different things could be studied better than when different versions of groups are tested separately as is done now.

We agree with the reviewer that the metformin use at conception could have a significant impact on s-PROK1, and we also considered two-way ANOVA as a statistical test. Unfortunately, the distribution of s-PROK1 after log-transformation was still not normal, therefore we chose Mann-Whitney U-test instead of two-way ANOVA.

3. I think also the effect of study should be accounted in the analyses especially when the intervention was not exactly the same in both studies.

Indeed, there was a slight difference between the two trials regarding the metformin dose, 1700 mg and 2000 mg daily, respectively, which might have affected our results. However, in an epi-analysis of these 2 RCTs, metformin treatment did not seem to affect the prevalence of gestational diabetes or pre-eclampsia, but significantly decreased late miscarriage/preterm delivery. Interestingly, in the RCT with the higher dose of metformin, the prevalence of none of these pregnancy complications were significantly affected by metformin use. Therefore, we thought it is reasonable to perform our analysis on these 2 RCTs together. Now we added this information as a limitation of the study.

4. What was pilot study made for? Is there some information that was used in the planning of PregMet-study or what was the idea of pilot study?

This post-hoc analysis included 2 randomized, placebo-controlled, double blinded studies, namely the Pilot and the PregMet1 studies. The original aim of the Pilot study was to investigate a possible effect of metformin on androgen levels in pregnant women with PCOS at 80% power to detect a 1.0 nmol/l difference in change of testosterone level between groups when assuming SD of 1.0 nmol/l. As the secondary measures of the Pilot study were pregnancy complications and outcomes, the sample size calculation in the PregMet1 study was based on the results of the Pilot study to detect a 25% difference in preterm deliveries and insulin requiring GDM at the power of 90%.

5. Statistical analyses: Where is log-transformed S-PROK1 used? Analyses were reported to be performed using non-parametric tests where normality is not assumed.

We log-transformed s-PROK1 to comfort the data to normality. As the log-transformed data still did not follow normal distribution, we chose to use the non-parametric Mann-Whitney U-test to compare s-PROK1 between different subgroups.

6. Results and tables: Exact p-values should be presented instead of only below or over 0.05.

Now exact p-values are presented in the tables, figures and in the Results of the manuscript.

7. Results presented in Figures 2 and 3 should be presented also more accurately with exact numbers. It is difficult to evaluate the difference between groups without exact values (with 95% confidence intervals if applicable). Also the clinical importance of difference could be discussed.

We log-transformed the data to decrease skewness towards large values. We tested plotting the raw data, however, we found that it made very difficult to interpret the results. We chose to plot the well justified log-transformed data to make patterns more interpretable and this is the data for which we

are answering questions about, also regarding identifying the outliers and extreme outliers.

8. Figure 1: Metformin use at conception would be good additional information in flow chart.

The information on metformin use at conception is added now to the flowchart.

9. Figure 2: It would be more informative to see one figure with four groups according to randomized group and metformin use at conception.

As we could not perform two-way ANOVA to analyze metformin use at conception and later during pregnancy based on randomization due to the lack of normality of the data, we think that it would be misleading to show all the 4 groups on the same figure.

10. Figures 2 and 3: Number of observations in each group should be presented in the figures.

The numbers of observations are now added to the figure legends.

VERSION 2 – REVIEW

REVIEWER	Hurme, Saija University of Turku, Department of Biostatistics
REVIEW RETURNED	12-Sep-2023

GENERAL COMMENTS	General comments: Article has improved a lot but I would still recommend to revise the methodology used to test differences in s-PROK1 in subgroups to make these results more clear for reader. Statistical analyses, Results, Figures and Tables: To clarify the article, I recommend to change additional subgroups to four groups according to randomized group and metformin use at conception (1: placebo and no Metformin use at conception, 2: placebo and Metformin use at conception, 3: Metformin and no Metformin use at conception, 4: Metformin and Metformin use at conception). Difference between those four groups can be tested also using non-parametric Kruskal-Wallis test. If p-value of this test is significant, then pairwise comparisons could be done using Mann-Whitney U - test and then for example Bonferroni method can be used to correct p-values because of multiple testing. Then also figures 1 a, c and d can be replaced with one figure and supplementary tables 1 b, c, and d can be replaced with one table. I think that these subgroups would be much more interesting and clear to the reader than confusing overlapping groups that are now reported. With these four groups it would also be easier to separate the effect of metformin at conception and during the pregnancy. Also it would be useful to see the medians of s-PROK1 with confidence intervals for the randomized groups and also the subgroups. Results: Page 7, Line 8-9: “independent of later randomization”. This phrase is used when the model also includes the variable of later randomization. I think this was not the situation and therefore this phrase “independent” can not be used here.
---

VERSION 2 – AUTHOR RESPONSE

Comment of the reviewer:

General comments: Article has improved a lot but I would still recommend to revise the methodology used to test differences in s-PROK1 in subgroups to make these results more clear for reader.

Comment:

Statistical analyses, Results, Figures and Tables: To clarify the article, I recommend to change additional subgroups to four groups according to randomized group and metformin use at conception (1: placebo and no Metformin use at conception, 2: placebo and Metformin use at conception, 3: Metformin and no Metformin use at conception, 4: Metformin and Metformin use at conception). Difference between those four groups can be tested also using non-parametric Kruskal-Wallis test. If p-value of this test is significant, then pairwise comparisons could be done using Mann-Whitney U - test and then for example Bonferroni method can be used to correct p-values because of multiple testing.

Answer:

We thank the reviewer for the valuable and constructive comments. We have now changed the statistical method from Mann-Whitney U-test to median test and the post hoc pairwise median test, as we feel that this test fits better with our skewed data. We have changed the Materials and Methods and the Results section accordingly.

Comment:

Then also figures 1 a, c and d can be replaced with one figure and supplementary tables 1 b, c, and d can be replaced with one table. I think that these subgroups would be much more interesting and clear to the reader than confusing overlapping groups that are now reported. With these four groups it would also be easier to separate the effect of metformin at conception and during the pregnancy.

Answer:

We now replaced Figure 2 c and d, showing 2 and 2 subgroups with Figure 3, showing four subgroups together. However, we have kept Figure 2a, as we think it is an important to compare metformin and no metformin use at conception regardless of later randomization. We also have kept Supplementary table 1, as we wished to show the baseline characteristics of subgroups that significantly differ from one another.

Comment:

Also it would be useful to see the medians of s-PROK1 with confidence intervals for the randomized groups and also the subgroups.

Answer:

Now we added the median and the 95% CI to all the groups and subgroups in the text.

Comment:

Page 7, Line 8-9: “independent of later randomization”. This phrase is used when the model also includes the variable of later randomization. I think this was not the situation and therefore this phrase “independent” can not be used here.

Answer:

We agree with the reviewer and therefore we changed “independent” to “regardless”.

VERSION 3 – REVIEW

REVIEWER	Hurme, Saija University of Turku, Department of Biostatistics
REVIEW RETURNED	24-Oct-2023

GENERAL COMMENTS	I'm now satisfied with the statistical methods and how Figure 3 is presented. I would still like to know the results (p-values and medians with 95% CI) of other pairwise comparisons in Figure 3 even though there is not statistically significant differences (as I assume the situation is).
--

VERSION 3 – AUTHOR RESPONSE

Reviewer: 4

Dr. Saija Hurme, University of Turku

Comments to the Author:

I'm now satisfied with the statistical methods and how Figure 3 is presented. I would still like to know the results (p-values and medians with 95% CI) of other pairwise comparisons in Figure 3 even though there is not statistically significant differences (as I assume the situation is).

Answer:

We thank the reviewer for the valuable and constructive comment. Originally, our post hoc pairwise median test included 4 comparisons (no metformin at conception/metformin vs metformin at conception/metformin; no metformin at conception/metformin vs no metformin at conception/placebo; metformin at conception/metformin vs no metformin at conception/placebo; no metformin at conception/placebo vs metformin at conception/placebo) from the 6 possible comparisons. In this revised version of the manuscript, we included 5 pairwise comparisons (addition of comparison metformin at conception/metformin vs metformin at conception/placebo), which slightly changed the p-values, without affecting the main outcome. As we were seeking information on the effect of metformin use either at conception and/or during the pregnancy on s-PROK1, we think that the comparison no metformin at conception/metformin vs metformin at conception/placebo is irrelevant, therefore we did not include this in the post hoc pairwise analysis in our work. We have changed the Results section and Figure 3 accordingly. We also added the median and 95% CI to all the groups in the text. For the sake of unity, we added the p-values to the non-significant results, too, in the text. For the sake of clarity, we show here how the p-values after performing the pairwise median test would look like if all the 6 possible comparisons would have been included:

no metformin at conception/metformin vs metformin at conception/metformin
p=0.008 padjusted=0.049

no metformin at conception/metformin vs no metformin at conception/placebo
p=0.6128 padjusted=1.000000

metformin at conception/metformin vs no metformin at conception/placebo
p=0.001088 padjusted=0.006528

no metformin at conception/placebo vs metformin at conception/placebo

p=0.5765 padjusted=1.000000

metformin at conception/metformin vs metformin at conception/placebo
p=0.06014 padjusted=0.3608006

no metformin at conception/metformin vs metformin at conception/placebo
p=0.6128 padjusted=1.000000